# An Automatic System for Continuous Pain Intensity Monitoring Based on Analyzing Data from Uni-, Bi-, and Multi-Modality

**DOI:** 10.3390/s22134992

**Published:** 2022-07-01

**Authors:** Ehsan Othman, Philipp Werner, Frerk Saxen, Marc-André Fiedler, Ayoub Al-Hamadi

**Affiliations:** Department of Neuro-Information Technology, Institute for Information Technology and Communications, Otto-von-Guericke University Magdeburg, 39106 Magdeburg, Germany; philipp.werner@ovgu.de (P.W.); frerk.saxen@ovgu.de (F.S.); marc-andre.fiedler@ovgu.de (M.-A.F.); ayoub.al-hamadi@ovgu.de (A.A.-H.)

**Keywords:** continuous pain intensity monitoring, facial expressions, electrocardiogram, electrodermal activity, electromyogram, fused modalities, random forest, long short-term memory network, sample weighting

## Abstract

Pain is a reliable indicator of health issues; it affects patients’ quality of life when not well managed. The current methods in the clinical application undergo biases and errors; moreover, such methods do not facilitate continuous pain monitoring. For this purpose, the recent methodologies in automatic pain assessment were introduced, which demonstrated the possibility for objectively and robustly measuring and monitoring pain when using behavioral cues and physiological signals. This paper focuses on introducing a reliable automatic system for continuous monitoring of pain intensity by analyzing behavioral cues, such as facial expressions and audio, and physiological signals, such as electrocardiogram (ECG), electromyogram (EMG), and electrodermal activity (EDA) from the X-ITE Pain Dataset. Several experiments were conducted with 11 datasets regarding classification and regression; these datasets were obtained from the database to reduce the impact of the imbalanced database problem. With each single modality (Uni-modality) experiment, we used a Random Forest [RF] baseline method, a Long Short-Term Memory (LSTM) method, and a LSTM using a sample weighting method (called LSTM-SW). Further, LSTM and LSTM-SW were used with fused modalities (two modalities = Bi-modality and all modalities = Multi-modality) experiments. Sample weighting was used to downweight misclassified samples during training to improve the performance. The experiments’ results confirmed that regression is better than classification with imbalanced datasets, EDA is the best single modality, and fused modalities improved the performance significantly over the single modality in 10 out of 11 datasets.

## 1. Introduction

Pain is an indicator of the human body’s health condition; it is a warning mechanism that gives a trusted message to indicate people to pay serious medical attention and respond quickly. Recent studies have shown that behavioral cues and physiological signals are very informative for pain detection [1,2,3]. Therefore, several automated methods have been proposed using these cues and signals to overcome the limitations of current clinical applications for pain recognition, such as their unreliability and non-continuous monitoring for patients, especially those who cannot self-report their pain (infants, intensive care patients, or people suffering from dementia). This paper focuses on introducing an automated system that can be used to complement the current clinical methods for better pain management by analyzing behavioral and physiological data from five modalities of the X-ITE Pain Dataset. This multimodal pain database provides additional valuable information for more advanced discriminating pain or pain intensities versus no pain because it includes different qualities and durations of pain [4].

Visual cues (facial expressions), vocalization cues (verbally and non-verbally), and physiological signals (bio-signals) express pain physically; these cues and signals play an important role in assessing pain in individuals [3]. Facial expressions are one of the more important cues in predicting pain based on analyzing human behaviors during pain [5,6]; across different nationalities, genders, cultures, ages, and genders, pain is expressed similarly. Further, some individuals express pain non-verbally by moaning, crying, groaning, and sighing (vocalization cues), and some express pain verbally by mentioning pain or using offensive words, and some express their pain [7,8]. Additionally, the physiological signals are informative indicators of pain, they cause alterations in tissues and organs (e.g., skin, heart, and muscles’ electrical properties). The most widely used signals are electrocardiogram (ECG), facial electromyography (EMG), and electrodermal activity (EDA) [9,10].

For objective pain intensity assessment, an increasing number of studies [10,11,12,13,14] has investigated behavioral cues and physiological signals with machine learning models. The findings demonstrated that the EDA modality mostly outperformed other physiological single modalities in terms of accurate pain assessment; EMG was the second best modality. Further, the temporal integration of extracted features from each modality improved the performance of pain assessment [14,15,16]. The temporal integration was represented for each modality by a time series statistics descriptor that was calculated from several statistical measures with their first and second derivatives per time series. The purpose of this paper is to confirm the above works’ findings, utilize temporal integration, and combine the best two modalities (EMG and EDA) and all modalities to improve the pain intensity monitoring performance; various studies [10,13,14,17,18] motivated us to use the combined modalities. The combination of both behavioral and physiological pain cues/signals is potentially good for developing objective pain assessment regarding the Odhner et al. [19] study. We also compared the results of the single modalities versus fused modalities (two and all modalities) to introduce the best automated system for objective continuous monitoring of pain intensity, which could be highly beneficial for reliable and economic pain intensity assessment.

The current work is organized as follows. Section 1.1 describes the contribution of this work. Section 2 provides an overview of pain recognition methods based on behavioral and physiological responses and describes their relevance to this paper. Section 3 shows a description of the materials and methods for automatic monitoring of continuous pain intensity using fused modalities: The X-ITE Pain Database preprocessing is shown in Section 3.1, automatic monitoring methods for continuous pain intensity regarding classification and regression are explained in Section 3.2, and the experimental setup is explained in Section 3.3. Section 4 presents a comparison between models’ results when using single and fused modalities regarding classification and regression, followed by a discussion in Section 5 and a conclusion in Section 6.

### 1.1. Contributions

Werner et al. [14] and Walter et al. [16] reported the results of using phasic (short) and tonic (long) stimulation samples from facial expressions, audio, and psychological data of 7 s, which were cut out from the continuous recording of the main stimulation phase in X-ITE Pain Database. This paper goes beyond their studies, as it reports the first continuous pain monitoring results based on analyzing data from the best two fused modalities (EMG and EDA) and all fused modalities (facial expressions, audio, ECG, EMG, EDA) of the same database. In line with this paper, the facial expressions modality results were reported in [20,21] and both the facial expressions and EDA modalities’ results were reported in [22]. The findings from this paper outperformed those from our recent paper [20,21,22]. We used the continuous recording of each modality of most of the data collection experiments, which is about 1 h and a half per subject. Further, we compared classification and regression results when using RF, LSTM, and LSTM-SW. We did not apply RF with fused modalities experiments because it was not needed due to its lower performance compared to LSTM and LSTM-SW during best single modality (EDA) experiments. We preferred to continue experiments with promising methods. Additionally, we compared models’ performances achieved using EDA (best single modality) with fused modalities (the best two modalities (EMG and EDA) and all modalities) to obtain the best method of recognizing pain intensity for the two pain stimulus types (phasic and tonic) in three pain intensities (low, moderate, and severe) and two qualities (heat and electrical stimuli). The findings of this work are the baseline results for future research related to real-time pain intensity monitoring systems using the EDA and fused modalities ((EMG and EDA) and all modalities) in the X-ITE Pain Database.

## 2. Related Work

Many studies have focused on various possible objective indicators for pain. The behavioral and physiological pain indicators are commonly used for pain assessment. Facial expressions and vocalizations (from audio) are considered behavioral pain indicators; heart rate variability, muscle activity, and electrodermal activity are considered physiological pain indicators. Recent studies reported that the fusion of these indicators increased the accuracy of distinguishing no pain versus pain [14,16,19]. Ekman and Friesen [23] decomposed facial expressions into individual facial Action Units (AUs) with the Facial Action Coding System (FACS). Examples of facial changes associated with pain are brow lower, cheek raise, lids tight, nose wrinkle, nasolabial deepen, upper lip raise, lip corner pull, lip stretch, lips apart, jaw drop, lids droop, eyes closed, and blink [1,23,24,25,26]. Vocalizations include verbal, non-verbal, and breathing behaviors. They define during pain experienced as the utterance of sounds, noises, and words using the vocal apparatus. Verbal vocalizations include protests and complaints by mentioning pain or using offensive words [27]; non-verbal vocalizations include moaning, crying, groaning, and gasps [7]; the changes in breathing patterns such as sighing have also been considered as vocalizations in regard to Waters et al. [28].

Clinical studies [29,30,31,32,33] have provided strong empirical evidence for the correlation between individual physiological signals and pain. The pain process starts from the sensory receptors (also called nociceptors) by noxious thermal, chemical, or mechanical stimuli, which can be activated to the body from an external or internal source. The information regarding detecting harmful stimuli and converting these into electrical signals is transduced via nociceptors and transmitted through the spinal cord to the brain. Then, specific parts of the brain are responsible for responding to pain signals, which are the prosencephalon, mesencephalon, and cortex [34,35,36,37]. ECG signals captures the changes in the electrical activity of the heartbeats (low-frequency/high-frequency ratio) and the heart rate interval. The Heart Rate Variability (HRV) is calculated on ECG data; the changes of HRV in the low-frequency power increase during painful stimulation [38,39]. EMG signals measures the changes in electrical properties of the muscle; EMG activity is often measured at the zygomaticus (mouth corner raiser), trapezius (back of the neck), and corrugator superscillii (brow lowerer) muscles [40]. EDA signals record the changes in the electrical activity of the skin when using two electrodes connected to the index and ring fingers. In response to a pain stimulus, EDA is a good measure because of intense body activity after experiencing pain; when a painful stimulus is applied, the sympathetic nervous system (SNS) activates the finger’s sweat glands to produce more sweat, and this, in turn, increases skin conductance [41,42,43].

Many promising methods were introduced for automatically recognizing pain using behavioral and physiological pain indicators because they carry valuable information about different pain levels in patients. The automatic pain recognition consists of three main steps: (1) face detection and registration/vocalizations preprocessing/physiological preprocessing, (2) feature extraction, and (3) pain recognition. Several automatic methods using facial expressions were introduced for recognizing frame-level and sequence-level pain intensity. Frame-level methods ignore temporal information and are thus limited in describing relevant dynamic information that is beneficial for pain intensity recognition. Further, occlusion, such as self-occlusion, oxygen mask, and pacifier, is another limitation of using such methods. Thus, many recent works focus on sequence-level pain recognition because it is more effective in describing such information [44,45,46]. According to the ability of RF [47] for pain detection using facial expressions, audio, or physiological indicators [13,14,20,21,22,45,48,49], we introduced in this work RF as a baseline method for single modalities experiments and compared its performance to the proposed deep learning methods (LSTM and LSTM-SW) that analyze data encoding temporal information.

Most methods of pain recognition used a single modality [44,50] used video, [51,52] used audio signal, and [10,12,33] used physiological signals. The recent methods used multiple modalities [3,4,14,16,53,54,55] that can improve the performance and flexibility of pain recognition. Thiam et al. [54] proposed multi-modal methods to develop pain intensity classification. They introduced a supervised deep learning method and a self-supervised method for recognizing pain intensity based on physiological signals. The self-supervised method automatically generated physiological data and simultaneously performed a fine-tuning of the deep learning model, which had been previously trained on a significantly smaller amount of data. Thus, they were able to significantly improve the data efficiency. Hinduja et al. [55] introduced a multi-modal method for pain recognition by fusing facial expressions and physiological signals (heart rate, respiration, blood pressure, and electrodermal activity) from the BP4D+ database. The fusion improved accuracy when evaluation included all subjects or same gender compared to using only one modality (facial expressions or physiological signals). Pouromran et al. [10] focused on pain intensity estimation using the BioVid dataset; they built different machine learning models for continuous pain estimation: Linear Regression, Support Vector Regression (SVR), Neural Networks, k-nearest neighbors, Random Forest, and XGBoost. They used the extracted features from a single modality and the combination features from multiple modalities. The EDA single model outperformed multi-modal results for pain intensity recognition, and SVR gave the best predictive performance across different sensors.

The current automated methods for pain assessment have not yielded estimation accuracies acceptable in clinical settings due to several reasons: (1) Half of the available pain databases for research purposes contain response data from a single modality (e.g., facial expressions). (2) The multimodal pain databases have a significant impact on the performance of automatic pain assessment systems; to the common belief that the quality and duration of pain are need to provide additional valuable information for more advanced discriminating pain or pain intensities versus no pain, the X-ITE Pain Database has been made to complement existing databases and the analysis of pain regarding quality and length. (3) There are some studies to assess pain reported promising results of using more than three modalities; they show that to obtain a reliable pain assessment system, it is recommended to analyze pain and detecting valid pain patterns from multiple modalities, including both behavioral cues and physiological signals. (4) Few of the current pain assessment methods focus on continuous monitoring, which is more necessary for pain assessment for prompt pain detection and immediate intervention. Therefore, this paper addresses those limitations and proposes an automatic and multimodal system for continuous monitoring of pain intensity using five sensor modalities (facial expressions, audio, ECG, EMG, and EDA) from the X-ITE Pain Database. A combination of behavioral cues and physiological signals was used with appropriate machine learning methods. LSTM [56] and LSTM using sample weighting (LSTM-SW) were utilized with a single or multiple fused modalities (facial expression, audio, ECG, EMG, EDA, (EMG and EDA), or all modalities) to predict continuous phasic or tonic pain intensity versus no pain. LSTM was proposed to learn long-term dependency among longer time periods by storing information from previous periods. The sample weighting method [48] was applied to reduce the weight of misclassified samples by duplicating some training samples with more facial responses if their classification scores were above 0.3 to improve the pain intensity recognition performance.

## 3. Materials and Methods

Figure 1 shows the methodology of the proposed automatic system for continuous monitoring of pain intensity. We pre-processed the input data from the five modalities (facial expressions, audio, ECG, EMG, and EDA) to extract useful features. Then, we calculated temporal integration features from time series data coming from those five modalities. After that, we prepared the experimental data by further processing the obtained data from the temporal integration process; such processing was suggested to overcome imbalanced database and outlier problems; for more details, see Section 3.1. Regarding classification and regression, three types of experiments were introduced to analyze the experimental data for continuous monitoring of pain intensity: (1) Uni-modality (data from single modalities) experiments, (2) Bi-modality (fusing data from two modalities) experiments, and (3) Multi-modality (fusing data from five modalities) experiments. We applied RF as baseline methods (Random Forest classifier (RFc) and Random Forest regression (RFr)) with single modalities experiments, Long Short-Term Memory (LSTM) and LSTM using the sample weighting method (LSTM-SW) with single and fused modalities experiments; see Section 3.2. For more details about the conducted experiments, see Section 3.3. 

### 3.1. X-ITE Pain Database Pre-Processing

In this work, a multimodal Experimentally Induced Thermal and Electrical (X-ITE) Pain Database [4] includes data that were recorded when healthy participants (subjects) were exposed to two different pain qualities (heat and electric) in three intensities (low, medium, and high) and two different pain stimuli durations (phasic (5 s = short) and tonic (1 min = long)). The subjects were aged between 18 and 50 years (at least 90% Caucasian participants = 67 men and 67 women). The heat pain stimulus was stimulated at participants’ forearm using a thermal stimulator (Medoc PATHWAY Model ATS, Medoc—Advanced Medical Systems, Ramat Yishay, Israel). The electrical pain stimulus was stimulated at participants when electrodes were attached to participants’ index and middle fingers using an electrical stimulator (Digitimer DS7A, Digitimer Ltd., Letchworth Garden City, UK). A person calibration was applied prior to the main stimulation phase experience, which means the participant self-reported the pain experienced during several stimuli using the numeric rating scale. Three intensities of both pain stimuli (heat and electricity) of each duration pain (phasic or tonic) were selected individually based on participants’ personal pain sensitivity (tolerances). For each phasic stimulus, the three pain intensities (times two pain modalities) were repeated 30 times for 5 s duration, applied in randomized order with pauses of 8–12 s. The 1-min tonic stimuli were applied once per intensity, followed by a pause of 5 min. The EDA includes both background tonic (Skin Conductance Level: SCL = 1-min pain stimuli) and rapid phasic components (Skin Conductance Responses: SCRs = 5-min pain stimuli). There were three phases of how tonic heat and electrical pain intensity stimuli were applied: the two lower intensities were applied randomly during the phasic stimulus period, and the highest intensity was applied at the end of the experiment. The entire experiment (preparation and actual experiment) took about 3 h per participant; approximately 1 and a half hours was the duration of the actual experiment for each participant, which was used in this work. For more details see Gruss et al. [4]. The five modalities were analyzed to objectively monitor the phasic and tonic pain intensity during the application of the thermal and electrical pain stimuli and no pain.

In line with [14,16,20,21,22,48], the subset of the same 127 human healthy participants (subjects) was selected, including samples only, for which data were available from five sensor modalities (frontal facial RGB video, audio, ECG, EMG, and EDA). The data from the five modalities from X-ITE Pain Database were pre-processed by (1) processing the frontal facial RGB video using OpenFace [57] for detecting the face from each frame for each participant (subject) and for extracting facial features (FF) and head pose, (2) processing the audio signal using openSMILE [58], (3) applying the QRS-detection algorithm by Hamilton et al. [59] with the ECG signals, (4) processing the three EMG channels with a zero-phase third-order Butterworth band-pass filter, and (5) keeping EDA without filtering. The FF that were used included 3 head poses (Yaw, Pitch, and Roll), AU1 (binary occurrence output), and 17 AU intensity outputs of OpenFace, which are AU1 (Inner Brow Raiser), AU2 (Outer Brow Raiser), AU4 (Brow Lowerer), AU5 (Upper Lid Raiser), AU6 (Cheek Raiser), AU7 (Lid Tightener), AU9 (Nose Wrinkler), AU10 (Upper Lip Raiser), AU12 (Lip Corner Puller), AU14 (Dimpler), AU15 (Lip Corner Depressor), AU17 (Chin Raiser), AU20 (Lip stretcher), AU23 (Lip Tightener), AU25 (Lips Part), AU26 (Jaw Drop), and AU45 (Blink). The low-level descriptor (LLD) was extracted from audio modality, comprising 4 energy features (sum of auditory spectrum (loudness), sum of RASTA-filtered auditory spectrum, root-mean-square (RMS) energy, and zero-crossing rate), 6 voicing features (F0 (Subharmonic Summation (SHS)\and Viterbi smoothing), probability of voicing, logarithmic harmonics-to-noise ratio (HNR), Jitter (local and delivered duty paid (DDP)), Shimmer (local)), and 14 spectral features (Mel Frequency Cepstral Coefficients (MFCCs)). With processed ECG, R-to-R intervals were used to determine heart rate. Afterward, we interpolated the heart rate signal linearly to match the sampling of the EMG and EDA (1000 Hz). The 21-dimensional facial expression/24-dimensional audio/1-dimensional ECG/3-dimensional EMG/1-dimensional EDA time series were selected at a frame rate of 25 frames per second (fps).

The temporal integration for each modality was represented by a time series statistics descriptor [44,60] to describe the changes of features from each modality, which are called Facial Activity Descriptor (FAD), Audio Descriptor (Audio-D), ECG Descriptor (ECG-D), EMG Descriptor (EMG-D), and EDA Descriptor (EDA-D). Each second in each descriptor was summarized by four statistics of the time series itself and its first and second derivative, including minimum, maximum, mean, and standard deviation, yielding a 12 × 21-dimensional, 12 × 24-dimensional, 12 × 1-dimensional, 12 × 3-dimensional, and 12 × 1-dimensional descriptor for facial, audio, ECG, EMG, and EDA features, respectively. A person-specific standardization of the features [44] was applied with all descriptors in order to focus on the within-subject response variation rather than the differences between subjects. For each subject, the mean and standard deviation were calculated, then each feature value was subtracted by the mean and divided by the standard deviation that belonged to the same subject. RF, LSTM, and LSTM-SW were used with all descriptors. The descriptor for each modality we use was at the same frame rate (1/25 s = 25 fps) in line with [20,21,22]. A sliding window strategy was used to obtain input samples with a time length of 10 s ago.

In the X-ITE Pain Database, −10 indicates samples with problems such as false start and restart of the stimuli, overlapping between heat or electrical stimulation, unbalanced phasic estimation, short pause, short tonic electrical stimulus, single heat stimulus in front, or additional stimulus. Furthermore, −11 indicates the samples when the subject speaks or interacts during the experiment (the beginning and after the first and second tonic stimuli of the experiment). The remaining annotations were presented in Table 1.

Several pre-processing steps were proposed on the X-ITE Pain Database to reduce the impact of the extremely imbalanced database problem: First, we investigated the intensities of facial expressions for most samples when expressing pain intensity, then we assigned all subjects into four categories based on how they expressed pain intensity. Second, we suggested splitting the database into three splits: training set, validation, and testing set. Randomly, subjects for each split from each category were selected based on 80% of data for training (100 subjects = 572,696 samples), 10% for validation (13 subjects = 75,537 samples), and 10% for testing (14 subjects = 79,485 samples). Each split contained subjects from all intensity categories. Third, the obtained splits from the database were processed: (a) all sequences of samples with labels −10, −11, and no pain samples sequence before and after these samples were excluded to simplify the problem and reduce the impact of imbalance in the database; (b) the obtained dataset was split into 6 subsets to evaluate the proposed methods (see the Subsets, which are the first six datasets in Table 2); (c) each obtained dataset was reduced by removing some no pain samples prior to pain intensity samples in a time series for each subject to evaluate the proposed methods; these datasets are called Reduced Subsets; see the Reduced Subsets, which were the last six datasets in Table 2.

### 3.2. Automatic Pain Intensity Monitoring Methods

Random Forest (RF) was used because it is an applicable method regarding classification and regression tasks. RF is parallelizable method, which means that the process can be split into multiple machines to run, and this leads to a faster computation time (faster to train and predict). In contrast, Boosting is a sequential ensemble method, which takes longer to compute. Further, RF is good with high dimensionality data, robust to outliers and non-linear data, good to handle imbalanced data, and it has also low bias and moderate variance. Alongside other studies [14,16,20,21,22], RFc and RFr were trained with 100 trees and a maximum depth of 10 nodes for classification and regression tasks with single modality experiments. The RFc method in [14,16,48] showed good results in predicting pain intensity and no pain from the time windows of samples that were cut out from the continuous recording of the main stimulation phase. LSTM is an effective method for better handling time series prediction compared to other time series methods because it has a memory cell that can maintain information in memory for long periods of time. It is more accurate on datasets using large sequences. LSTM classification architecture comprised a single LSTM layer activated by ReLU followed by a flatten layer, and then one dense layer activated by ReLU. The final output layer had *n* neurons (*n* = number of classes). The Softmax was used as the activation function in the output layer, and the Categorical Cross-Entropy (CCE) was the use as the loss function. The configurations of LSTM regression architecture were similar to LSTM classification architecture, except the final output layer with one neuron was activated using a sigmoid function. The pre-processed data (time series data = samples) were inserted into LSTMs one by one in sequence.

After observing the highly imbalanced database and imbalanced proposed datasets (see Table 1 and Table 2), LSTM again was used after increasing the weight of the training samples with more facial responses, called LSTM using sample weighting (LSTM-SW). The sample weighting method was based on duplicating some samples with high scores. The samples with prediction scores higher than 0.3 in training data when using the RFc with FAD modality were determined, and then these samples were replicated once. The duplicates are desirable, as some single images could appear multiple times per epoch because the LSTM model puts more weight on getting these samples (with observable pain reaction) correct and focuses less on samples without an observable pain reaction. The samples after increasing were trained on the LSTM for classification and regression. To ensure comparability of test results, samples were never duplicated in the test data.

### 3.3. Experiments

Three categories of experiments applied on 11 datasets from the X-ITE Pain Database: Uni-modality experiments using data from single modalities, Bi-modality experiments using data from two modalities, and Multi-modality experiments using data from multiple modalities. RF was used only with single modality (Uni-moality) experiments. Due to RF’s poor performance compared to LSTM and LSTM, we decided to continue two and all fused modalities experiments using LSTM and LSTM. For more details about those methods, see Section 3.2. For reference, a Trivial classifier was calculated within classification experiments, which always votes for the majority class of the dataset (no pain in our experiments). In order to be able to know which modality is best for monitoring continuous pain intensity, the suggested automatic methods were trained with the time series data from each single modality. In these experiments, the time series data (experimental data = FAD, Audio-D, ECG-D, EMG-D, and EDA-D) were used individually to predict pain intensity using RF, LSTM, and LSTM-SW for classification (discrete predictions) and regression (continuous predictions). All LSTM/LSTM-SW classification models were optimized using the loss function CCE, and LSTM/LSTM-SW regression models were optimized using the loss function BCE. The obtained models were trained for 2000 epochs with different lr $\left(10^{-4}\;\mathrm{or}\;10^{-5}\;\mathrm{or}\;10^{-6}\right)$ when setting up the batch size equal to 512 and using adam optimizer. Table 3 shows the architecture of LSTM/LSTM-SW within Uni-modality experiments.

The architectures A(c), B(c), C(c), and D(c) for classification and A(r), B(r) for regression all have input size of 10 × 252/288/36/12, the number of features was variant according to the used modality. The number 10 indicates timesteps (25 Hz time series were reduced to one Hz after applying temporal integration process), 252 indicates facial features (FAD), 288 indicates audio features (Audio-D), 12 indicates ECG features (ECG-D) or EDA features (EDA-D), and 36 indicates features (EMD-D). A(c) and C(c) classification architectures comprised a single LSTM layer with 4 units activated by ReLU followed by a flatten layer, and then one dense layer with 128 neurons activated by ReLU. The final output layer had 7 neurons in A(c) and 4 neurons in C(c). B(c) and D(c) classification architectures comprised a single LSTM layer with 8 units activated by ReLU and followed by flatten layer, and then one dense layer with 64 neurons activated by ReLU. The final dense output layer had 7 neurons in B(c) and 4 neurons in D(c). The Softmax was used as the activation function in the output layer, and the CCE was the use as the loss function. The configurations of A(r) regression architecture were similar to A(c) and C(c), and the configurations of B(r) regression architecture were similar to B(c) and D(c), except the final output layer with one neuron was activated using a sigmoid function. 

Two Uni-modality architectures (FAD/EMG-D and EDA-D) were combined using LSTM/LSTM-SW by merging their final dense layers using a concatenate layer (see Table 4). A-Bi(c) and C-Bi(c) classification architectures of (FAD and EDA-D) Bi-modality or (EMG-D and EDA-D) Bi-modality, including A(c) and C(c), comprised a single LSTM layer with 4 units activated by ReLU and followed by a flatten layer, and then one dense layer with 128 neurons activated by ReLU. The output of dense layer (dense1) from X (EDA-D) modality architecture was concatenated with the output of dense layer (dense1) from Y (FAD/EMG-D) modality. The final layer (dense2) had 7 neurons in A-Bi(c) and 4 neurons in C-Bi(c). B-Bi(c) and D-Bi(c) classification architectures, including B(c) and D(c), comprised a single LSTM layer with 8 units activated by ReLU and followed by flatten layer, and then one dense layer with 64 neurons activated by ReLU. The output of dense layer (dense1) from X (EDA-D) modality architecture was concatenated with the output of dense layer (dense1) from Y (FAD/EMG-D) modality. The final layer (dense2) had 7 neurons in B-Bi(c) and 4 neurons in D-Bi(c). The configurations of A-Bi(r) regression architecture were similar to A-Bi(c) and C-Bi(c), and the configurations of B-Bi(r) regression architecture were similar to B-Bi(c) and D-Bi(c), except the output layer with one neuron.

Table 5 shows the overview of the Multi-modality experiments. All single modalities’ architectures using LSTM/LSTM-SW were combined by concatenating the outputs from the dense1 layers using concatenate layer. The Multi-modality experiments were similar to Bi-modality experiments, except the input were the time series data from all modalities (FAD, Audio-D, ECG-D, EMG-D, and EDA-D). A-Mu(c) and C-Mu(c) classification architectures of Multi-modality, including A(c) and C(c), comprised a single LSTM layer with 4 units activated by ReLU and followed by a flatten layer, and then one dense layer with 128 neurons activated by ReLU. The dense2 layer had 7 neurons in A-Mu(c) and 4 neurons in C-Mu(c). B-Mu(c) and D-Mu(c) classification architectures, including B(c) and D(c), comprised a single LSTM layer with 8 units activated by ReLU and followed by flatten layer, and then one dense layer with 64 neurons activated by ReLU. The dense2 layer has 7 neurons in B-Mu(c) and 4 neurons in D-Mu(c). The configurations of A-Mu(r) regression architecture were similar to A-Mu(c) and C-Mu(c), and the configurations of B-Mu(r) regression architecture were similar to B-Mu(c) and D-Mu(c), except for the final dense out-put layer with one neuron.

The learning rate (lr) within A(c), A(r), A-Bi(c), A-Bi(r), A-Mu(c), A-Mu(r), C(c), C(r), C-Bi(c), C-Bi(r), C-Mu(c), and C-Mu(r) using PD, HPD, and EPD is $10^{-5}$; using TD, HTD, and ETD it is $10^{-6}$; and using RPD, RHPD, and REPD it is $10^{-4}$. The lr within B(c), B(r), B-Bi(c), B-Bi(r), B-Mu(c), B-Mu(r), D(c), D(r), D-Bi(c), D-Bi(r), D-Mu(c), and D-Mu(r) using RTD and RETD is $10^{-6}$. In the output layers, 7 levels indicate no pain versus heat and electrical pain intensity recognition for one of two different (phasic/tonic), and 4 levels indicate no pain versus heat/electrical pain intensity recognition for one of two different (phasic/tonic).

## 4. Results

In this section, we introduce the comparison between the proposed methods (RF, LSTM, and LSTM-SW) with the best single modality (EDA), best two fused modalities (EMG and EDA), and fused all modalities (facial expressions, audio, ECG, EMG, and EDA). The experimental data was used (EDA-D, (EMG-D and EDA-D), (FAD, Audio-D, ECG-D, EMG-D, and EDA-D)). Section 4.1 shows the findings when evaluating the test set using regression measures (Mean Squared Error (MSE) and Intraclass Correlation Coefficient (ICC) [61]), whereas Section 4.2 shows the findings in terms of classification measures when classification outperformed the regression in terms of regression measures. The classification measures were Accuracy, Micro average precision (Micro avg. precision), Micro average recall (Micro avg. recall), and Micro average F1-score (Micro avg. F1-score).

### 4.1. Classification vs. Regression

After comparing between single modalities models when applying baseline methods (RFc and RFr), LSTM, and LSTM-SW with 11 datasets from the experimental data, we found that EDA-D Uni-modality modality models performed the best with most datasets regarding classification and regression. Table 6 and Table 7 show the comparison of the best Uni-modality, the best Bi-modality, and Multi-modality models in terms of MSE and ICC measures, respectively. EMG-D outperformed EDA-D only with the TD dataset and performed similarly to EDA-D with the HTD dataset regarding classification. In line with Uni-modality modality experiments, the best results were obtained when using EMG-D and EDA-D Bi-modality with the same methods and all datasets except the HTD dataset, and FAD and EDA-D Bi-modality were the best. 

Regarding regression, both FAD and EDA-D Bi- and Multi-modality models performed similarly when using LSTM with the EPD dataset. The Multi-modality models were the best when using TD, ETD, and RETD; they got MSE of 0.08, 0.06, and 0.09, and the ICC of 0.30, 0.33, and 0.56, respectively. Further, both LSTM and LSTM-SW Uni-modality models outperformed the baseline Uni-modality model (RFc and RFr). Bi- and Multi-modality models outperformed Uni-modality models except with the REPD dataset, and EDA-D Uni-modality model when applying LSTM-SW was the best regarding regression (MSE of 0.03 and ICC of 0.88). The results did not improve after fusing EDA-D as the best Uni-modality with the other Uni-modality, possibly because the outlier rate was increased when using more than one modality. The EMG-D and EDA-D Bi-modality models performed the best with the PD, EPD, RHPD, RPD, and RTD datasets regarding regression. With HTD, Bi-modality models performed the best when applying LSTM-SW (MSE of 0.13 and ICC of 0.42) Classification performed better than regression with almost balanced datasets (HTD). Classification and regression performed similarly when using LSTM-SW and LSTM with HPD; they got MSE of 0.09 and 0.07, and ICC of 0.41 and 0.41, respectively. Figure 2 shows the best models after comparing between Uni-modality modality models (EDA-D modality), the best Bi-modality models (EMG-D and EDA-D Bi-modality), and Multi-modality models. Finally, almost all the Bi- and Multi-modality models performed the best regarding regression in terms of standard deviation (STD) and mean.

### 4.2. Classification

Classification models got the superior performance to regression models with the HTD dataset and performed similarly with the HPD dataset in terms of MSE and ICC measures. Thus, we applied RFc (EDA Uni-modality), LSTM, and LSTM-SW with HPD and HTD datasets regarding classification to introduce the best performance. Table 8 shows that LSTM and LSTM-SW Uni-modality models with HPD and HTD successfully predicted discrete pain intensity levels in sequences compared to Trivial and RFc in terms of Accuracy, Micro avg. precision, Micro avg. recall, and Micro avg. F1-Score.

The EMG-D and EDA-D Bi-modality and Multi-modality models when applying LSTM-SW with HPD and HTD were better than those models using LSTM in terms of Micro avg. recall and Micro avg. F1-Score. The *t*-test as inferential statistic used to determine if there is a significant difference between the means of two methods (RF and LSTM/LSTM-SW). Further, EMG-D and EDA-D Bi-modality models when applying LSTM-SW outperformed LSTM models with HTD in terms of Accuracy (49.8%) and Micro avg. precision (48.7%). EDA-D Uni-modality model, when applying LSTM-SW with HTD, performed the best in terms of Micro avg. recall (100%); however, the Bi-modality model when applying LSTM-SW with the same dataset performed excellent (97%). Additionally, the Multi-modality model, when applying LSTM-SW with HPD, performed the best in terms of Micro avg. recall (22.3%). In terms of F1-score, Bi-modality models, when applying LSTM-SW with HPD and HTD, performed the best, at about 26.3% and 63.3%.

After closely investigating the correctly predicted samples from classification results when applying RFc (EDA Uni-modality), LSTM, and LSTM-SW, we found that the models with HPD performed the best in recognizing no pain and the highest pain intensity, whereas models with HTD performed the best in recognizing intermediate pain (low and moderate); see Table 9. The reason is that models may have difficulty in recognizing intermediate pain intensity stimulation in large imbalanced datasets like HPD when HTD is less imbalanced (no pain (about 20%), intermediate pain (about 27%), and the highest pain (about 26%)).

Additionally, Bi- and Multi-modality models improved the performance of Uni-modality models when applying both LSTM and LSTM-SW with the imbalanced HPD. Finally, the EMG-D and EDA-D Bi-modality models, when applying LSTM-SW with HPD and HTD, performed the best based on calculating the average of the model performance using HPD and HTD, which are 37.5% and 48.2%, respectively.

## 5. Discussion

We conducted several experiments using three methods (RF, LSTM, and LSTM-SW) regarding classification and regression in order to gain insights into monitoring continuous pain intensity. Further, these experiments applied to compare the performance of proposed methods when using single modalities (using RF, LSTM, and LSTM-SW), two fused modalities, and all modalities (using LSTM and LSTM-SW) from the X-ITE Pain Database. The data from each modality were split into 11 datasets to simplify the imbalanced database problem, improve the results, and generalize the capability of the desired system. An additional reason for using these different datasets was to explore the generalization capability of a continuous pain intensity monitoring system. For evaluation, MSE and ICC were used to measure the performance of classification models versus regression models, and the best classification models outperformed the regression models were further evaluated using Accuracy, Micro avg. precision, Micro avg. recall, and Micro avg. F1-score. In this paper, we introduced the best results from single modality (EDA-D Uni-modality) experiments, two fused modalities (EMG-D and EDA-D Bi-modality) experiments, and all modalities (Multi-modality) experiments. The obtained results were alongside to prior works on the X-ITE Pain Database [14,16], the BioVid, and SenseEmotion databases [45,62,63]. EMG-D Uni-modality performed similarly to EDA-D Uni-modality with the almost balanced dataset (HTD); the changes in muscle activity (EMG) to heat tonic stimuli tend to be similar intense as the changes in the superficial muscles of the skin of the hand (EDA).

As we saw in the quantitative results in Section 3, the reduction strategy reduced the influence of outliers (unimportant samples) by reducing some no pain samples prior to pain intensity sequences in a time series for each subject; the Subsets performances improved after applying the reduction strategy (see Reduced Subsets results in Table 6 and Table 7 and Figure 2). It is important to reduce the noise in imbalanced data than to keep hard samples. Further, regression outperforms classification when using huge imbalanced datasets, whereas classification was the best with the almost balanced dataset (HTD); see Figure 2. Both classification and regression perform similarly with Heat Phasic Dataset (HPD) in terms of ICC measure. Trivial failed to recognize pain intensity with HTD and HPD, whereas the proposed methods are significantly better in terms of Micro avg. precision, Micro avg. recall, and Micro avg. F1-score; see Table 8. Accuracy measure does not a proper measure with imbalanced datasets, because it does not distinguish between the numbers of correctly classified examples of different classes. After investigating the HPD and HTD datasets further, we assumed that classification performed well when the distribution of pain intensity samples was high in the imbalanced datasets (the majority of samples experienced no pain), which was 26.7% with HTD, followed by HPD (7.71%) based on the mean value; see Table 9. It shows also LSTM-SW is the best compared to LSTM when fusing two modalities of EMG and EDA (EMG-D and EDA-D Bi-modality) regarding classification with HPD and HTD datasets. Additionally, LSTM-SW increased the performance compared to LSTM of several datasets; see Table 6 and Table 7. The success of LSTM-SW is based on downweighted samples in the training set with a less facial response by using the sample weighting method; these samples might negatively affect the model performance.

Models using the best two and all fused modalities (EMG-D and EDA-D Bi-modality and Multi-modality) improved the continuous monitoring of pain intensity significantly compared to those using the best single modality (EDA-D Uni-modality) in 10 out of 11 datasets. Only the performance of the EDA-D Uni-modality model did not improve when using the REPD dataset, which might be due to the conflicts between modalities in these datasets, especially because of the outliers. In line with [14,16], Multi-modality models outperformed EDA-D Uni-modality models and EMG-D and EDA-D Bi-modality models with imbalanced tonic datasets (TD, ETD, and RETD). The possible reason is that the responses to tonic stimuli are more intense than the response to phasic stimuli for each modality. Further, the response to electrical tonic stimuli tends to start earlier and more rapidly than the response to heat for each modality. Thus, all modalities could significantly benefit from this Multi-modality model when the responses in all modalities are more intense. With EPD, both EMG-D and EDA-D Bi-modality and Multi-modality models performed the same. The EMG-D and EDA-D Bi-modality models outperformed Multi-modality models when using LSTM and LSTM-SW on six datasets: PD, HPD, HTD, RPD, RHPD, and RTD; see the acronyms description in Table 2, probably because the outliers in the worst modalities (audio and ECG). The ECG is sensitive to miscellaneous mixed noises. The audio signal includes many label noises, possibly due to the vocalizations responses when some subjects were stimulated with different pain intensities or when some subjects talk or produce other vocal responses during no pain is experienced. A reliable and trustworthy pain assessment is necessary for adequate pain management, changing analgesic dose, and additional interventions if required. Additionally, good care requires more than one pain intensity measure (self-report, external observations, or physiological tests. Therefore, this work was introduced as a baseline study for future research regarding continuous pain intensity monitoring systems using single or multiple modalities with the X-ITE pain database. Further, we recommend using our data pre-processing strategy and the proposed methods with other pain databases; considering the reliable results in this paper, they will probably perform well.

## 6. Conclusions

As seen in Figure 3, the proposed system confirms that it is possible to monitor continuous pain intensity using machine learning models with fused modalities (facial expressions, audio, and physiological (ECG, EMG, and EDA)). The difficulties for any model are the huge imbalance in the database and the outliers. The models failed to recognize minority classes and deal with noise when using the database. Due to the distribution of the classes being variants, we split the database into six datasets in terms of four qualities of pain stimuli (phasic (short) and tonic (long) variants of each heat and electrical stimuli). Further, we reduced the noise in the modalities data by removing some no pain samples prior to pain intensity samples in a time series for each subject in each of the six datasets. There are plenty of no pain labels, which are inconsistent with the modalities’ responses. After applying different models with all datasets, we concluded that regression is better than classification with imbalanced datasets, which is the case with real data. Regression reduces the effect of confounding variables by isolating the effect of each variable by allowing for the role of each independent variable to learn without worrying about other variables in the model. Further, LSTM and LSTM-SW successfully recognize pain intensity compared to RF (used with single modalities). We also confirmed that EDA is the best single modality, and EMG and EDA are the best two fused modalities. Two and all fused modalities improved results further compared to a single modality with almost all datasets. The fused modalities that use all modalities were the best with imbalanced tonic datasets.

EMG and EDA fused modality performed the best with imbalanced phasic datasets and balanced dataset (HTD); this result shows that EMG and EDA are very good options for cost-effective pain monitoring; there is no need to use all modalities. However, there is some limitation of using this system: (1) The database does not contain any of a vulnerable group; however, this system can help to predict pain, particularly with vulnerable patients, but it has not yet been implemented for them. (2) Although applying the proposed reduction strategy to the imbalanced datasets helps to improve the performance, there are still plenty of outliers that limit further performance improvement. (3) A larger dataset with more pain intensities is necessary for more reliable automatic monitoring of continuous pain intensity.

## Figures and Tables

**Figure 1 sensors-22-04992-f001:**
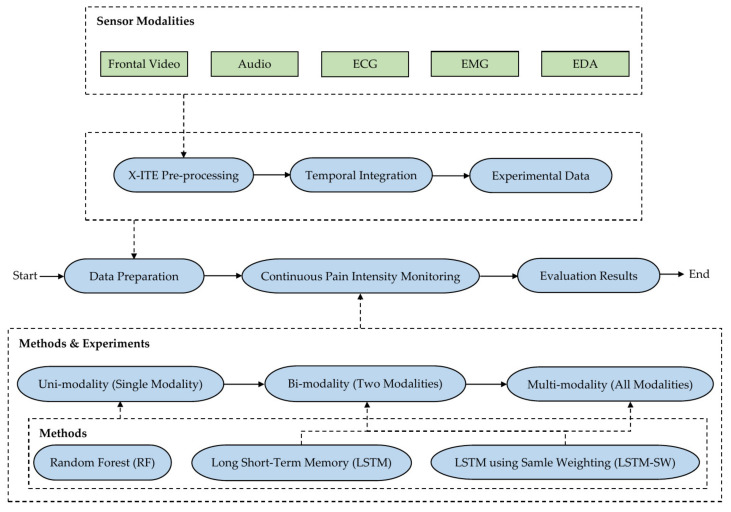
The general pipeline of the proposed automatic system for continuous monitoring of pain intensity.

**Figure 2 sensors-22-04992-f002:**
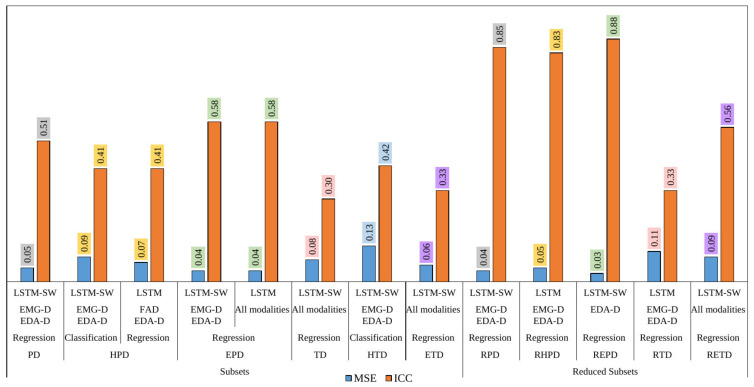
The best results of applying Uni-, Bi, and Multi-modality using RF, LSTM, and LSTM-SW regarding classification and regression in terms of MSE and ICC measures. Numbers with the same background color were used to easily highlight the improvement in the performance of each dataset before and after the reduction strategy was applied.

**Figure 3 sensors-22-04992-f003:**
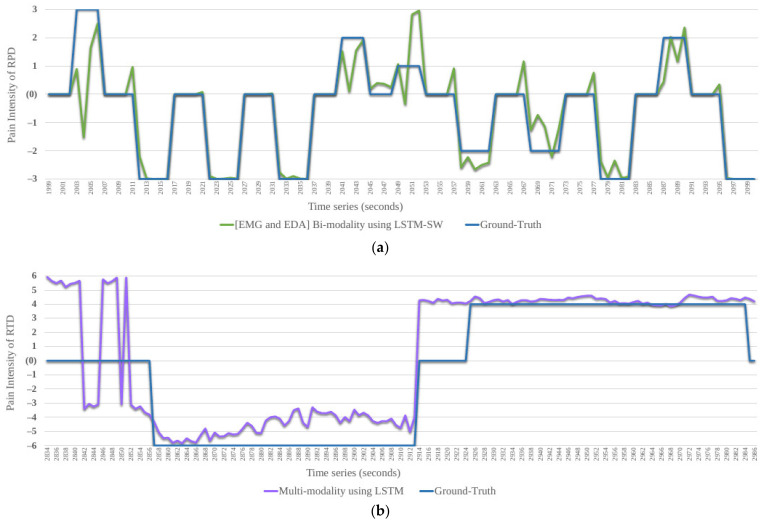
A performance example of the proposed automated system for continuous monitoring pain intensity from test set, including results of the best models with RPD (see (**a**)) and RTD (see (**b**)) compared to Ground−Truth. RPD: Reduced Phasic Dataset. RTD: Reduced Tonic Dataset.

**Table 1 sensors-22-04992-t001:** List abbreviations of pain stimuli type, modalities, intensities, and numerical class labels with the percentage samples distribution. This table is also presented in our recent study [20,21,22].

Type	Modality	Intensities			
		Severe	Moderate	Low	No Pain (77%)
Phasic	H	PH3 = 3 (2%)	PH2 = 2 (2.1%)	PH1 = 1 (2.1%)	BL = 0
	E	PE3 = −3 (2.6%)	PE2 = −2 (2.6%)	PE1 = −1 (2.6%)	
Tonic	H	TH3 = 6 (1%)	TH2 = 5 (1%)	TH1 = 4 (1%)	BL = 0
	E	TE3 = −6 (1%)	TE2 = −5 (1%)	TE1 = −4 (1%)	

E: Electrical pain stimulus, H: Heat pain stimulus, BL: Baseline; −10 and −11 labels not used in the experiments: −10 (0.5%) and −11(2.5%).

**Table 2 sensors-22-04992-t002:** List of datasets with samples’ distribution based on labels.

Subsets (Experimental Data)			No Pain	Pain Intensities
PD	Phasic Dataset	Exclude tonic samples and no pain samples before these samples and also after samples with −10, −11 labeled.	77.7%	22.23%
HPD	Heat Phasic Dataset	Exclude electrical samples from PD and no-pain samples before these frames.	87.5%	21.5%
EPD	Electrical Phasic Dataset	Exclude heat samples from PD and no pain frames before these frames.	86.1%	13.9%
TD	Tonic Dataset	Exclude phasic samples and no pain samples before these samples and also after samples with −10, −11 labeled.	70.3%	29.7%
HTD	Heat Tonic Dataset	Exclude electrical samples from TD and no pain frames before these frames.	20.0%	80.0%
ETD	Electrical Tonic Dataset	Exclude heat samples from TD and no pain frames before these frames.	82.0%	18.0%
**Reduced Subsets (Experimental Data)**			**No Pain**	**Pain Intensities**
RPD	Reduced Phasic Dataset	Reduce the no pain frames in PD to about 50%.	50.0%	50.0%
RHPD	Reduced Heat Phasic Dataset	Reduce the no pain frames in HPD to about 50%.	50.1%	49.9%
REPD	Reduced Electrical Phasic Dataset	Reduce the no pain frames in EPD to about 50%.	50.0%	50.0%
RTD	Reduced Tonic Dataset	Reduce the no pain frames in TD to about 38%.	38.1%	61.9%
RETD	Reduced Electrical Tonic Dataset	Reduce the no pain frames in ETD to about 49%.	49.0%	51.0%

**Table 3 sensors-22-04992-t003:** A summary of the LSTM/LSTM-SW architectures’ configurations using data from Uni-modality. This table is also presented in our recent study [20,21,22].

Layer Type	Attribute	Classification				Regression	
		A(c)	B(c)	C(c)	D(c)	A(r)	B(r)
Input	Size:	10 × 252	10 × 252	10 × 252	10 × 252	10 × 252	10 × 252
	Timestep:	10	10	10	10	10	10
	Features:	252	252	252	252	252	252
LSTM	Activation:	ReLU	ReLU	ReLU	ReLU	ReLU	ReLU
	No. of units:	4	8	4	8	4	8
Dropout	with p:	0.5	0.5	0.5	0.5	0.5	0.5
Flatten	Output:	80	40	80	40	80	40
Dense1	Activation:	ReLU	ReLU	ReLU	ReLU	ReLU	ReLU
	No. of units:	128	64	128	64	128	64
Dense2	Activation:	Softmax	Softmax	Softmax	Softmax	Linear/Sigmoid	
	No. of units:	7	7	4	4	1	1
Output	Continuous	-	-	-	-	√	√
	Discrete	√	√	√	√	-	-
		7 levels	7 levels	4 levels	4 levels		

**Table 4 sensors-22-04992-t004:** A summary of the LSTM/LSTM-SW architectures’ configurations using data from FAD/EMG-D and EDA-D Bi-modality. A(c), B(c), C(c), D(c), A(r), and B(r) are LSTM/LSTM-SW architectures in Table 3.

Layer Type	Attribute	Architectures Configurations (Bi-Modality)					
		Classification				Regression	
		A-Bi(c)	B-Bi(c)	C-Bi(c)	D-Bi(c)	A-Bi(r)	B-Bi(r)
Concatenate (after dense1)	Modality X	A(c)	B(c)	C(c)	D(c)	A(r)	B(r)
	+	+	+	+	+	+	+
	Modality Y	A(c)	B(c)	C(c)	D(c)	A(r)	B(r)
Dense2	Activation:	Softmax				Sigmoid	
	No. of units:	7	7	4	4	1	1
Output	Continuous	-	-	-	-	√	√
	Discrete	√	√	√	√	-	-
		7 levels	7 levels	4 levels	4 levels		

**Table 5 sensors-22-04992-t005:** A summary of the LSTM/LSTM-SW architectures’ configurations using Multi-modality. A(c), B(c), C(c), D(c), A(r), and B(r) are LSTM/ LSTM-SW architectures using Uni-modality (see Table 3).

Layer Type	Attribute	Architectures Configurations (Multi-Modality)					
		Classification				Regression	
		A-Mu(c)	B-Mu(c)	C-Mu(c)	D-Mu(c)	A-Mu(r)	B-Mu(r)
Concatenate (after dense1)	Modality 1	A(c)	B(c)	C(c)	D(c)	A(r)	B(r)
	+	+	+	+	+	+	+
	Modality 2	A(c)	B(c)	C(c)	D(c)	A(r)	B(r)
	+	+	+	+	+	+	+
	Modality 3	A(c)	B(c)	C(c)	D(c)	A(r)	B(r)
	+	+	+	+	+	+	+
	Modality 4	A(c)	B(c)	C(c)	D(c)	A(r)	B(r)
	+	+	+	+	+	+	+
	Modality 5	A(c)	B(c)	C(c)	D(c)	A(r)	B(r)
Dense2	Activation:	Softmax	Softmax	Softmax	Softmax	Sigmoid	Sigmoid
	No. of units:	7	7	4	4	1	1
Output	Continuous	-	-	-	-	√	√
	Discrete	√	√	√	√	-	-
		7 levels	7 levels	4 levels	4 levels		

**Table 6 sensors-22-04992-t006:** Comparison of the best Uni-, Bi-, and Multi-modality models regarding classification and regression tasks with MSE measure. Meas.: Measure. The cells with light grey background indicate the models using LSTM, and cells with pink background indicate the models using LSTM-SW. The bold font indicates the best results.

Meas.	Task		Classification			Regression		
	Dataset		Uni-Modality	Bi-Modality	Multi-Modality	Uni-Modality	Bi-Modality	Multi-Modality
**MSE**	Subsets	PD	0.09 EDA-D	0.08 EMG-D EDA-D	**0.08**	0.06 EDA-D	**0.06** **EMG-D** **EDA-D**	0.06
		HPD	0.10 EDA-D	**0.09** **EMG-D** **EDA-D**	0.10	0.08 EDA-D	**0.07** **EMG-D** **EDA-D**	0.09
		EPD	0.06 EDA-D	0.06 EMG-D EDA-D	**0.05**	0.05 EDA-D	**0.04** **EMG-D** **EDA-D**	0.04
		TD	0.11 EDA-D	0.11 EMG-D EDA-D	**0.11**	0.09 EDA-D	0.10 EMG-D EDA-D	**0.08**
		HTD	0.15 EDA-D (LSTM-SW) EMG-D (LSTM)	**0.13** **EMG-D** **EDA-D**	0.15	0.11 EDA-D	0.10 EMG-D EDA-D	**0.10**
		ETD	0.11 (RFc)	0.08 EMG-D EDA-D	**0.08**	0.07 EDA-D	0.06 EMG-D EDA-D	**0.06**
	STD		0.03	0.02	0.03	0.02	0.02	0.02
	Mean		0.10	0.09	0.10	0.08	0.07	0.07
	Reduced Subsets	RPD	**0.05** **EDA-D** **(both LSTM)**	**0.05** **EMG-D** **EDA-D**	0.05	0.04 EDA-D	**0.04** **EMG-D** **EDA-D**	0.04
		RHPD	0.07 EDA-D	**0.07** **EMG-D** **EDA-D**	0.08	0.05 EDA-D (both LSTM)	**0.05** **EMG-D** **EDA-D**	0.08
		REPD	0.05 EDA-D (both LSTM)	**0.05** **EMG-D** **EDA-D** **(both LSTM)**	0.05 (both LSTM)	**0.03** **EDA-D**	0.04 EMG-D EDA-D	0.06
		RTD	0.19 EDA-D	**0.19** **EMG-D** **EDA-D**	0.19	0.11 EDA-D	**0.11** **EMG-D** **EDA-D**	0.04
		RETD	0.16 EDA-D	**0.15** **EMG-D** **EDA-D**	0.16	0.10 EDA-D	0.09 EMG-D EDA-D (both LSTM)	**0.09**
	STD		0.07	0.06	0.07	0.04	0.03	0.02
	Mean		0.10	0.10	0.11	0.07	0.07	0.05

**Table 7 sensors-22-04992-t007:** Comparison of the best Uni-, Bi-, and Multi-modality models regarding classification and regression tasks with ICC measure. Meas.: Measure. The cells with light grey background indicate the models using LSTM, and cells with pink background indicate the models using LSTM-SW. The bold font indicates the best results.

Meas.	Task		Classification			Regression		
	Dataset		Uni-Modality	Bi-Modality	Multi-Modality	Uni-Modality	Bi-Modality	Multi-Modality
**ICC**	Subsets	PD	0.40 EDA-D	0.45 EMG-D EDA-D	**0.46**	0.43 EDA-D	**0.51** **EMG-D** **EDA-D**	0.49
		HPD	0.30 EDA-D	**0.41** **EMG-D** **EDA-D**	0.39	0.32 EDA-D	**0.41** **EMG-D** **EDA-D**	0.40
		EPD	0.50 EDA-D	0.53 EMG-D EDA-D	**0.57**	0.53 EDA-D	**0.58** **EMG-D** **EDA-D**	**0.58**
		TD	0.15 EDA-D	0.18 EMG-D EDA-D	**0.23**	0.17 EDA-D	0.26 EMG-D EDA-D	**0.30**
		HTD	0.33 EDA-D (LSTM-SW) EMG-D (LSTM)	**0.42** **EMG-D** **EDA-D**	0.35	0.30 EDA-D	0.32 EMG-D EDA-D	**0.38**
		ETD	0.14 (RFc)	0.22 EMG-D EDA-D	**0.26**	0.21 EDA-D	0.31 EMG-D EDA-D	**0.33**
	STD		0.14	0.14	0.13	0.13	0.13	0.10
	Mean		0.30	0.39	0.38	0.33	0.40	0.41
	Reduced Subsets	RPD	**0.83** **EDA-D** **(both LSTM)**	**0.83** **EMG-D** **EDA-D**	0.82	0.84 EDA-D	**0.85** **EMG-D** **EDA-D**	0.82
		RHPD	0.76 EDA-D	**0.79** **EMG-D** **EDA-D**	0.74	0.81 EDA-D (both LSTM)	**0.83** **EMG-D** **EDA-D**	0.73
		REPD	0.84 EDA-D (both LSTM)	**0.85** **EMG-D** **EDA-D** **(both LSTM)**	0.81 (both LSTM)	**0.88** **EDA-D**	0.87 EMG-D EDA-D	0.80
		RTD	0.31 EDA-D	**0.32** **EMG-D** **EDA-D**	0.28	0.24 (EDA-D)	**0.33** **EMG-D** **EDA-D**	0.29
		RETD	0.47 EDA-D	**0.52** **EMG-D** **EDA-D**	0.44	0.49 EDA-D	0.52 EMG-D EDA-D (both LSTM)	**0.56**
	STD		0.24	0.23	0.24	0.28	0.24	0.22
	Mean		0.64	0.66	0.62	0.65	0.75	0.62

**Table 8 sensors-22-04992-t008:** Comparison of the best Uni-, Bi-, and Multi-modality models with HTD regarding classification task. The Bi-modality models. * *p* < 0.05 when using paired *t*-test between RFc and LSTMs (LSTM and LSTM-SW). The bold font indicates the best results.

Measure	Datasets	HPD				HTD			
	Model	Triv.	RFc	LSTM	LSTM-SW	Triv.	RFc	LSTM	LSTM-SW
Accuracy %	EDA-D (Uni-modality)	78.5	78.1	79.8 *	79 *	20	41.0	48.4	47.7
	FAD and EDA-D (Bi-modality)	78.5	-	**80.5 ***	80.2 *	20	-	47.4 *	**49.8 ***
	Multi-modality	78.5	-	79.3	77.6	20	-	41.6	42.2
Micro avg. precision%	EDA-D (Uni-modality)	0	24.6	36.6 *	32.2 *	0	42.7	48.2	47.7
	FAD and EDA-D (Bi-modality)	0	-	**42.8**	40.1 *	0	-	47.2	48.7
	Multi-modality	0	-	34.9 *	29.2	0	-	41.8	42.0
Micro avg. recall%	EDA-D (Uni-modality)	0	3.4	9.9 *	10.9 *	0	71.0	94.6 *	**100 ***
	FAD and EDA-D (Bi-modality)	0	-	16.3 *	21.4 *	0	-	92.9 *	97 *
	Multi-modality	0	-	19.8 *	22.3 *	0	-	90.8 *	**99.9 ***
Micro avg. F1-Score%	EDA-D (Uni-modality)	0	5.9	15.2 *	15.5 *	0	52.9	62.3 *	62.5 *
	FAD and EDA-D (Bi-modality)	0	-	22.3 *	**26.3 ***	0	-	60.7 *	63.3 *
	Multi-modality	0	-	23.6 *	24 *	0	-	56	57.9 *

**Table 9 sensors-22-04992-t009:** Recall% result of 4-Class continuous pain intensity recognition tasks of HPD and HTD on testing set. Uni-modality refers to EDA-D Uni-modality, Bi-modality refers to EMG and EDA-D Bi-modality. The bold font indicates the best results.

Model		Dataset									
		HPD					HPD				
		BL	PH1	PH2	PH3	Mean	BL	TH1	TH2	TH3	Mean
EDA-D (Uni-modality)	Trivial	100	0	0	0	25	100	0	0	0	25
	RFc	98.7	1.5	2	6.3	27.1	34.4	27.3	47	54.5	40.8
	LSTM	99.3	5.2	5.4	15.2	31.3	9.4	73	35.1	67.8	46.3
	LSTM-SW	98	3.1	1.7	23.3	31.5	0.30	72	39.8	68	45
FAD and EDA-D (Bi-modality)	Trivial	100	0	0	0	25	100	0	0	0	25
	RFc	-	-	-	-	-	-	-	-	-	-
	LSTM	98.7	6.5	5.2	29.5	35	20.3	45.4	56.1	62.5	46.1
	LSTM-SW	97.4	7.4	7.2	37.9	**37.5**	18.6	45.8	62.7	65.5	**48.2**
Multi-modality	Trivial	100	0	0	0	25	100	0	0	0	25
	RFc	-	-	-	-	-	-	-	-	-	-
	LSTM	96.8	6.2	4.6	36.2	36	11.2	56.3	28.9	63.5	40
	LSTM-SW	94.1	5.9	6.3	39.5	36.5	2.7	40.4	55.2	61.8	40

## Data Availability

Data are available from the authors upon request (to Sascha Gruss sascha.gruss@uni-ulm.de or Steffen Walter steffen.walter@uni-ulm.de) for researchers of academic institutes who meet the criteria for access to the confidential data.

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
