# Peer review of "An Automatic System for Continuous Pain Intensity Monitoring Based on Analyzing Data from Uni-, Bi-, and Multi-Modality"

_sensors, 2022, doi:10.3390/s22134992_

Round 1
Reviewer 1 Report
In this manuscript, the authors proposed a Continuous pain monitoring model with uni-, bi-, and multi-modality experiments. I have several comments.
1. classification and regression algorithm: This continuous pain monitoring model was proposed for clinical application. However, only the technical findings were discussed but not the significantly contributing components. For example, in the case of facial expression, what are those action units represent (movement, emotion, etc.), and which AU was more significantly contributing to the classification and regression? As the author indicated, SVR has been used for continuous pain estimation by Pouromran et al. SVM (linear kernel for classification or regression) provides interpretable clues. Thus, it would be a reasonable model to be included in the experiment.
2. modality: Tracy et al. (below) showed a relationship between HF HRV and pain, which represents participants’ autonomic state clinically than heart rate. Instead of testing 1 feature (HR), it would be informative to include HF HRV as the second feature for the ECG. It is not clear, for EDA, if the feature is skin conductance level (SCL, tonic) or skin conductance response (SCR, phasic) to the 5-sec or 1-min stimuli. EDA is influenced by the environmental temperature during the measurement. Was the room temperature controlled during the long data acquisition time? And African-Americans tend to have less SCR compared to other ethnic groups. Please provide demographics, if possible.
Tracy LM, Ioannou L, Baker KS, Gibson SJ, Georgiou-Karistianis N, Giummarra MJ. Meta-analytic evidence for decreased heart rate variability in chronic pain implicating parasympathetic nervous system dysregulation. Pain. 2016 Jan;157(1):7-29. doi: 10.1097/j.pain.0000000000000360. PMID: 26431423.
3. pain ratings: Participants’ pain levels were tested during the calibration period, if I understood correctly, and used for classification and regression. What do the 7-levels and 4-levels mean in Tables 3 and 4? Usually, participants’ pain ratings/levels change over time due to habituation or sensitization. It would be great if their pain ratings were collected in real-time. This (habituation/sensitization to thermal/electrical stimuli) is an important point for pain-related research.
4. main text: Some paragraphs of the manuscript (for example, lines 233-262) are redundant and not well organized to explain the protocol. An explanation for modality (for example, lines 206-207) is repeated several times throughout the manuscript.
Minor:
BL has never been defined.
Author Response
See the attachment, please.

Reviewer 2 Report
The authors conducted several experiments applied to 11 datasets from the X-ITE Pain Database using three methods (RF, LSTM, and LSTM-SW) regarding classification and regression to monitor continuous pain. The authors concluded that regression is better than classification with imbalanced datasets.
The authors provided details on the methods and results.
Here are my suggestions:
There are too many abbreviations which made the paper so difficult to read. Especially some of them were never explained, such as MSE and ICC.
It is better to include notes of the abbreviations used in the tables (e.g., Tables 6-9). The reader has to navigate the article to get the meanings of those abbreviations.
For the results in Table 8, if the comparisons were among RFc vs. LSTM vs. LSTM-SW, should you run repeated measured ANOVA instead of paired t-tests? It would be better to include the variances (e.g., standard deviations) and the means of the measures. I am also concerned about the potential of inflation of type I errors as so many paired tests were performed. You may want to discuss why paired t-tests were appropriate.
In Table 9, what are the meanings of those numbers? This table is not clearly interpreted.
In the discussion session, it would be better to discuss whether the methods for pre-processing the X-ITE Pain Database can be used for other databases.
There is also a lack of discussion on the computational requirements for each algorithm applied. What will be the ultimate goal for the automation pain assessment system? EDA is the best single modality, and EDA and EMG are the best two fused modalities were demonstrated but is this true for only pains from this specific database? Why is EDA the best for a single modality? It would be better to discuss the physiology of pain with those findings.
Author Response
See the attachment, please.

Round 2
Reviewer 1 Report
All comments were addressed.